# Impact of COVID-19 Pandemic on the Educational-Instructional Process of the Students from Technical Faculties

Elisabeta Spunei [1], Nătălița-Mihaela Frumușanu [2,*], Roxana Muntean [3] and Gabriela Mărginean [4]

1 Department of Engineering Sciences, Faculty of Engineering, Babes-Bolyai University, 400028 Cluj-Napoca, Romania; elisabeta.spunei@ubbcluj.ro
2 Department of Business Administration, Faculty of Economics and Business Administration, Babes-Bolyai University, 400028 Cluj-Napoca, Romania
3 Faculty of Mechanical Engineering, Politehnica University Timisoara, 300223 Timisoara, Romania; roxana.muntean@upt.ro
4 Westphalian Energy Institute, Westphalian University of Applied Sciences, 45897 Gelsenkirchen, Germany; gabriela.marginean@w-hs.de
* Correspondence: natalita.frumusanu@ubbcluj.ro; Tel.: +40-0724-089870

**Abstract:** Even though we live in a period when the word *digitization* is prevalent in many social areas, the COVID-19 pandemic has divided mankind into two main categories: some people have seen this crisis as an opportunity to move the activities online and, furthermore, to accelerate digitization in as many areas as possible, while others have been reluctant, keeping their preferences for face-to-face activities. The current work presents the results of an analysis on 249 students from 11 engineering faculties. The study aims to identify the impact of the COVID-19 pandemic on students' educational experiences when switching from face-to-face to online education during a public health emergency or COVID-19-related state of alert. The overall conclusion was that, although the pandemic has brought adverse consequences on the health and life quality of many people, the challenges that humankind has been subjected to have led to personal and professional development and have opened up new perspectives for carrying out the everyday activities.

**Keywords:** education; skills; competences; pandemic; online or face-to-face

## 1. Introduction

The rapid spread of human coronavirus has significantly changed human behaviour across the planet. Starting from an uncertain epidemiological situation in one of the most populous cities in Central China, the pandemic has quickly spread worldwide, generating a global crisis. Healthcare institutions and many other sectors, such as education, business, and tourism, have been severely impacted by the COVID-19 pandemic in several countries around the world [1]. The implementation of a social distancing protocol to prevent the person-to-person spread of coronavirus at the end of March 2020 has led to a rapid transition to an online educational-instructional process for most educational institutions worldwide [2].

In many cases, the pandemic had a generally negative impact, especially for some educational institutions, which had to quickly adapt to the new way of carrying out their main activities [3]. Closing some important institutions for a long period has challenged the education system, especially in countries with poor internet connection, or where students do not have access to such infrastructure [4]. This adjustment has affected both students and teachers, as well as the entire educational system that has had to ensure the efficient transfer from face-to-face to online activities. According to a literature review on the impact of the COVID-19 pandemic on worldwide educational activities, it has been concluded that online teaching and learning has various limitations, such as "*the weakness of online teaching infrastructure, the limited exposure of teachers to online teaching, the information gap,*

*non-conducive environment for learning at home, equity and academic excellence in terms of higher education"* [5].

The current study was performed in order to identify the impact of the COVID-19 pandemic on the educational-instructional process of students from technical faculties.

The research objectives were:

(a) To assess the impact of the measures applied to prevent the spread of COVID-19 on the educational activities carried out and the corresponding experienced effects (a–d);

(b) To identify the students' adaptability to change regarding their way of working, studying, and teaching using digital channels in the online environment (e,f);

(c) To evaluate the effectiveness of information communication through the act of teaching and students' need for interaction on an educational and personal level (g,k).

The objectives of the study were set in accordance with similar research carried out up to the date of the questionnaire, especially on the specifics of the teaching process in the technical field. The covered items and the answer variants included in the questionnaire were formulated in collaboration with several teachers and trainers who have dealt with a variety of challenging circumstances since switching to the online system.

The covered items are as follows:

(a) To indicate the impact level of the measures taken to prevent the spread of COVID-19 on the carried-out educational activities and the corresponding experienced effects. From the list created by the authors, the respondents were able to select one or more causes that affected their work. The causes listed in the questionnaire were:

- time spent in work teams;
- loss of contact with colleagues;
- reduced free time;
- lack of direct communication with teachers;
- stress caused by many online activities;
- difficult/insufficient understanding of the transmitted information/knowledge;
- insufficient training regarding equipment identification, measurement, testing skills;
- difficulty in processing experimental results;
- other.

(b) To identify the impact type (positive or negative) produced by moving the activities online and the opportunities identified by this measure. For this objective, respondents had the option to select *yes* or *no* to the question of whether the legislative restrictive measures taken to minimize the effects of COVID-19 were supportive for the activities' continuity, or to identify an opportunity determined by these measures;

(c) To specify the impact of the restrictive measures during the pandemic on professional training, where students could answer *yes* or *no*, and to mention positive and negative effects, respectively;

(d) To estimate the effect of self-isolation measures on time allocated to individual study;

(e) To evaluate the involvement of students in online activities and degree of participation in course or applicative activities (laboratory, project, seminar), respectively. For this objective, students had the opportunity to answer two questions, namely, whether they participated in online activities, with four possible answers (yes, yes partially, no, and no interest) and the degree of participation in these activities, also with four possible answers (participation between 0–25%, 25–50%, 50–75% or 75–100%);

(f) To indicate students' options regarding the teaching activities' approaches (students could choose between face-to-face teaching activities, online teaching activities, written material, other) and the type of assessment in the online system (multiple choice test, written-online applications, written-theoretical questions, oral, individual projects, team projects). Suggestions concerning the number of assessments for a subject and the percentage of the activity during the semester on the overall result of the subject were also analysed;

(g) To evaluate the possibility of replacing face-to-face activities by online activities, as well as the students' point of view regarding the opportunity of maintaining online activities to some extent;

(h) To indicate the students' degree of satisfaction concerning the online activities (low, medium, high) and students' perception on the possibility of increasing the quality of teaching activities through online activities (low, medium, high);

(i) To mention the elements considered pleasant in online learning, where students had the possibility to choose one of the options indicated by the authors (flexibility, various working tools, accessibility, digital skills, clear tasks), as well as to identify other facilities provided during courses. Furthermore, the main impediments of online learning were analysed. Thus, students had the option to choose one of the difficulties indicated by the authors in the questionnaire or to enter other personally identified barriers;

(j) To specify the elements on which the training of future engineers should focus on, be it digital competences, theoretical knowledge, or both;

(k) To suggest how to improve the online teaching process and further recommendations or support relevant to the online activities.

The paper is structured into four sections, which present the research methodology, the statistical analysis of the received answers and the results, summarized in form of the positive and negative effects of the teaching process, and the discussion section.

## 2. Materials and Methods

In order to identify the impact of the COVID-19 pandemic on the educational-instructional process of the students from technical faculties, from the trainees' perspective, a research study was initiated between March 2021 and July 2021. The research that has been carried out so far and which is reported in the literature provides general information, without specifically analysing the impact of e-learning on the teaching process in the technical field, where application activities in specialised laboratories must be predominant.

According to the literature, the data collection method is often chosen by each researcher based on the information needs correlated with the objectives pursued in the research. Therefore, in the current study, the questionnaire survey was selected, since it is considered to be one of the most widely used and efficient data collection methods. When designing and carrying out questionnaires, the following design recommendations are taken into account: setting the questions (statements); selecting the form/type of questions (open or closed); defining the structure of the questions and the format of the answer; determining the sequence of questions; determining the graphic layout of the questionnaire. Initially a pre-test was taken by students from Babeș-Bolyai University of Cluj; afterwards, it was distributed online to students from several university centres (Romanian, Moldova, and Germany). Before the online distribution, some modifications were made regarding the clarity and comprehensibility of the questions, since they were translated into English and German. In the end, the structure of the questionnaire was complex, comprising both closed-ended questions with single or multiple-choice answers and scale-based questions, where respondents were asked to rate the proposed criteria on a Likert scale from 1 to 5. During the distribution phase, specific activities were carried out, such as controlling the questionnaire distribution by the participating teachers and monitoring the response rate. Given the way the questionnaire was distributed, no reminder letters were sent regarding the deadline for questionnaire completion.

After closing the questionnaire developed in Google Forms, the Spreadsheet tool was used, which allowed to centralize all the answers offered by the students. Based on the obtained answers, different graphics were created in order to highlight the findings. For the open questions, the answers were roughly organised in similar thematic groups to emphasise the main positive and negative aspects identified by the respondents during the online activities.

Since the curriculum of technical faculties involves several application activities that call for the use of specific equipment, to which students did not have access during

online activities, the fundamental field of engineering was chosen for this study, rather than a specific university. The study consisted of an online questionnaire addressed to 224 students from 9 technical faculties in Romania (Reșița, Craiova, Hunedoara, Cluj, Timișoara, Iași, Galați, Baia Mare, Suceava) and one university from the Republic of Moldova, and 25 students from a technical university from Germany. The research aimed to identify the impact felt by students when switching from face-to-face to online activities during the COVID-19 pandemic. The questionnaire was distributed through university staff and collaborators from other university centres. Although there was not a mandatory activity for the students to complete the questionnaire, they chose to do so because they were curious to observe and learn about the difficulties they faced during the pandemic.

The students contributing to the current study belong to years 1–4 of bachelor and master cycles, and the questions did not analyse their background, social situation, or other criteria. No personal information of the students was requested in the questionnaire, therefore the participants were assured of the confidentiality of their answers. Since the applied questionnaire was anonymous and the respondent cannot be identified, the data gathered are not regarded as personal information. Anonymization is irreversible.

In addition, details about the respondents' educational background (bachelor or masters), gender, and the online platforms they used for their online activity were requested. The survey was developed using Google Forms, which allows its users to conduct surveys and compile the results in an Excel spreadsheet. The authors have access to these specific tools under an institutional licence. The questionnaire was distributed through teachers' communication channels and was open for completion throughout the period. No reminders were set during the data collection period.

## 3. Statistical Analysis

According to the results, among the 249 student respondents, 84.78% were undergraduate students and 15.22% were master students. By gender, 69.88% of the respondents declared their gender as male and 30.12% as female. The most utilized online platforms during the activities were Teams (51.81%) and Zoom (36.14%).

In order to achieve the objectives proposed in this paper, a qualitative analysis was carried out before the questionnaire was developed in order to formulate the response variables. This analysis was carried out by the authors in focus group discussions (focus groups being the seminar groups where we carried out our work).

Given that the questionnaire survey was selected as the data collection method for this research, as it is considered one of the most widely used and effective data collection methods, the analysis of the data obtained was quantitative. The data collected in excel format were filtered according to certain variables, and relationships and trends were identified in the data collected. Concerning the impact level of the measures taken to prevent the spread of COVID-19, most students concluded that the educational activity carried out was affected to a very great extent (39.76%) and to a great extent (33.33%). Figure 1 illustrates the students' responses concerning the causes that affected their educational activity during the pandemic. It can be observed that most students lacked socialization due to loss of contact with colleagues (65.1%) and a lack of direct communication with teachers (58.63%).

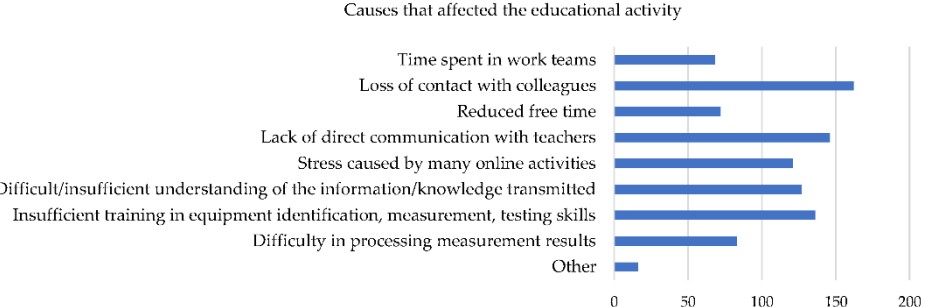

**Figure 1.** Causes that affected the educational activity during the COVID-19 pandemic.

Other causes identified by students experiencing a negative impact were related to the teaching activity, namely:

- insufficient training regarding equipment identification, measurement, and testing skills (54.62%);
- difficult/insufficient understanding of transmitted information/knowledge (51%);
- stress caused by many online activities (48.59%);
- difficulty in processing measurement results (33.33%).

The last ranked causes were related to time management:

- reduced free time—28.92%;
- time spent in work teams—27.31%.

In the category of other causes were identified:

- distracted by problems at home;
- too many activities/tasks;
- physical and mental health impairment caused by a high number of hours spent in front of the computer.

When it came to the influence that the shift to online teaching activities had on the students, it was discovered that 35.83% of the students felt positively about the change, compared to 64.17% of the students who felt negatively about it. Among the opportunities identified by this transition were:

- preventing the spread of the virus;
- increased study time due to elimination of travel time;
- use of simulation software that replaced lab activities on old stands;
- becoming more accustomed to using online learning platforms.

Concerning the impact of social distancing during the pandemic on students' professional training, 57% of the respondents did not feel that they were affected (Figure 2).

The impact of restrictive measures during the pandemic
on professional training

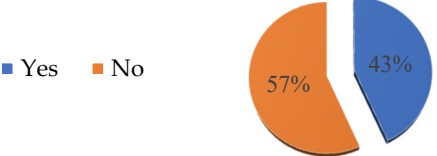

**Figure 2.** The impact degree of restrictive measures during the pandemic on professional training of the students.

Concerning students' involvement in online activities, from their answers, it can be concluded that 72% of the respondents participated in all of the activities they were informed about (Figure 3), and 25% partially participated due to difficulty accessing computer or internet services. There is a small percentage of students who did not have access to a computer/internet (1%) or were not interested (2%). Although the percentage of participants in online activities can be considered satisfactory, in some cases, this participation was not an active one. Regarding the degree of participation in online activities, 70% of respondents claimed that they took part in more than 75% of the scheduled activities.

Analysing the students' answers regarding the impact of self-isolation measures on the time allocated to individual study, there is a certain equality between the percentages for the three given situations (Figure 4). The majority of the respondents reported spending more time on individual study than usual, which can be attributed to the higher number of assignments received and the need for individual study due to the lack of additional support from teachers.

Analysing the students' reactions regarding the possibility that face-to-face activity might be replaced by online activity, it can be mentioned that 47% of the respondents

expressed that online activity should be complementary to face-to-face activity (Figure 5). This opportunity is also confirmed by the responses offered by the students concerning the suitability of maintaining online activities after the end of the pandemic, where 76% of respondents consider this action beneficial (Figure 6). There is also a slight hesitation for online studying, probably due to reluctance to change.

Involvement of students in online activities

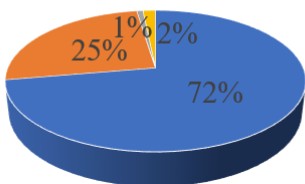

- Participate in all activities I have been informed about

- Partially participate, difficult access to computer or internet

- No, no acces to a computer or internet

- No, not interested

**Figure 3.** Students' involvement in online activities and degree of participation in course or applicative activities.

The impact of self-isolation measures on the time allocated to individual study

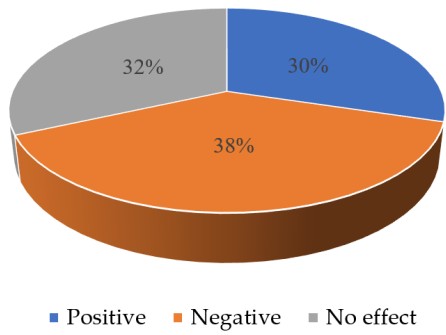

- Positive  - Negative  - No effect

**Figure 4.** The impact of self-isolation measures on the time allocated to individual study.

Can be the face to face activity replaced by the online activity ?

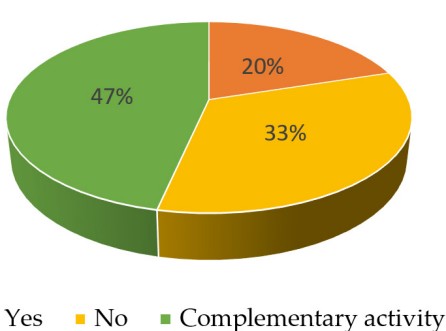

- Yes  - No  - Complementary activity

**Figure 5.** The possibility that face-to-face activity can be replaced by online activity.

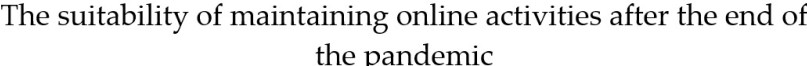

The suitability of maintaining online activities after the end of the pandemic

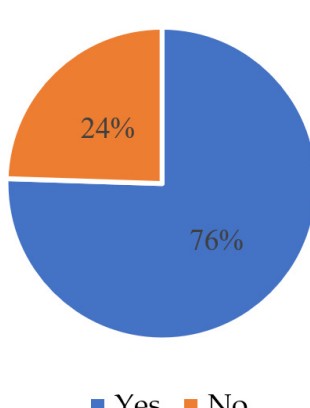

**Figure 6.** The suitability of maintaining online activities after the end of the pandemic.

The students' preferences on how to perform the teaching activities were found to be as follows: 54.62% of the respondents prefer face to face activity (Figure 7), while 32.13% prefer online activity. In the category of other modalities, all 17 respondents (6.83%) suggested online courses and face-to-face application activities.

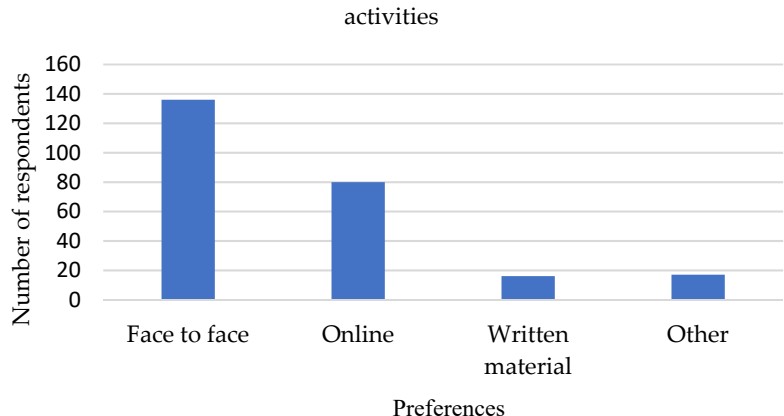

**Figure 7.** The students' preferences on how to perform the teaching activities.

According to the analysis of the data shown in Figure 8, the majority of students (58%) prefer multiple choice questions as a form of assessment, while 64.32% consider it appropriate to focus on one or a few forms of assessment, rather than combining as many forms of assessment as possible (35.68%).

When students were asked about the importance of the work accomplished during the semester in the final grade of a specific subject, 43% of respondents declared that the percentage should be between 41% and 50% (Figure 9), followed by 21–30% (22% of respondents) and 31–40% (17% of respondents).

Regarding the degree of satisfaction offered by the online activities, most of the respondents said that they were satisfied to a great extent (48.39%) and to an average extent (33.47%). The same perception is identified concerning the increase in the quality of the educational act through online activities (Figure 10), where most of the students believed it could be achieved to a great extent (39%), respectively, and to an average extent (38%).

Students' preferences regarding the online evaluation

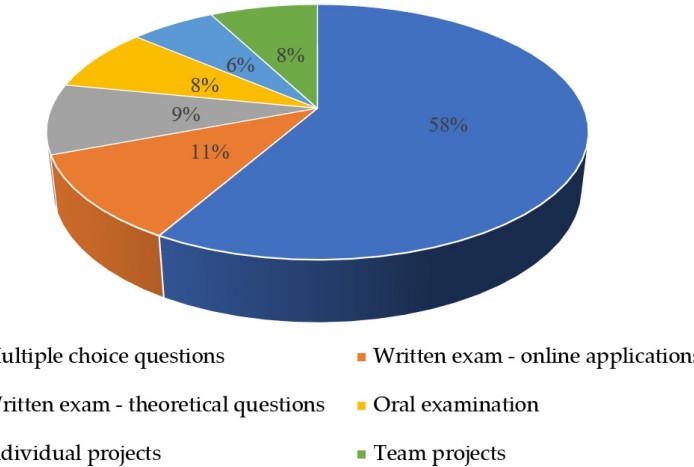

■ Multiple choice questions
■ Written exam - online applications
■ Written exam - theoretical questions
■ Oral examination
■ Individual projects
■ Team projects

**Figure 8.** Students' preferences regarding the online evaluation.

The importance of work accomplished during the semester in the final grade of subject

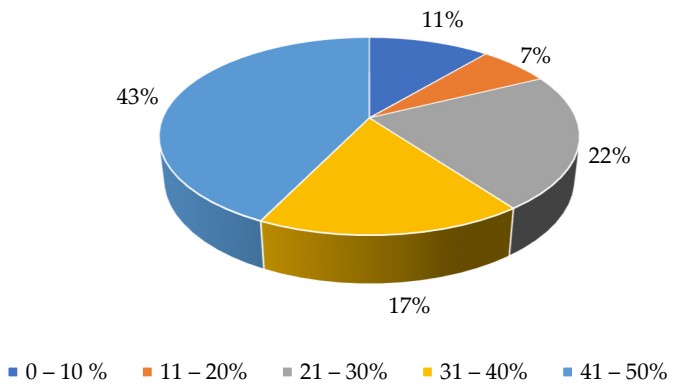

■ 0 – 10 %    ■ 11 – 20%    ■ 21 – 30%    ■ 31 – 40%    ■ 41 – 50%

**Figure 9.** The importance of work accomplished during the semester in the final grade for a specific subject.

Perception on the possibility of increasing the quality of the educational act through online activities

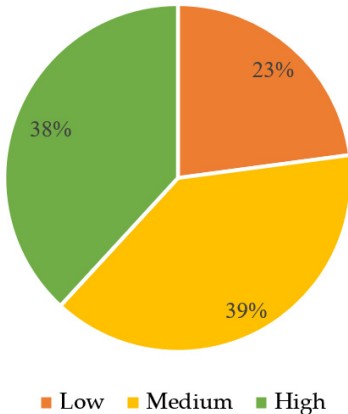

■ Low    ■ Medium    ■ High

**Figure 10.** Perception on the possibility of increasing the quality of the educational act through online activities.

Figure 11 indicates that the online activities surprised students, particularly because of the flexibility of activities (36%), accessibility to platforms/materials/resources (22%), and opportunity to acquire more digital skills (16%).

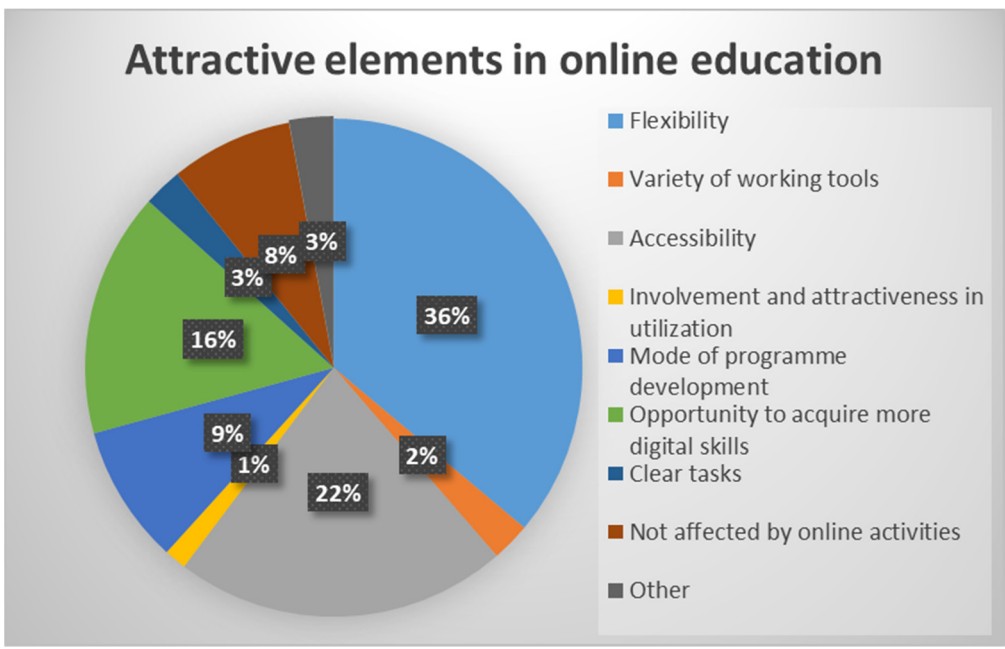

**Figure 11.** Attractive elements in online education.

The main obstacles students faced in online activities were according to Figure 12: access to technology (24%), limitations due to subject specifics (15%), modification of activities and content (14%), and technical difficulties (13%). In the category of other obstacles, 2% of respondents identified insufficient online education skills and a lack of necessary teacher competence as well as the outdated attitudes of both students and teachers.

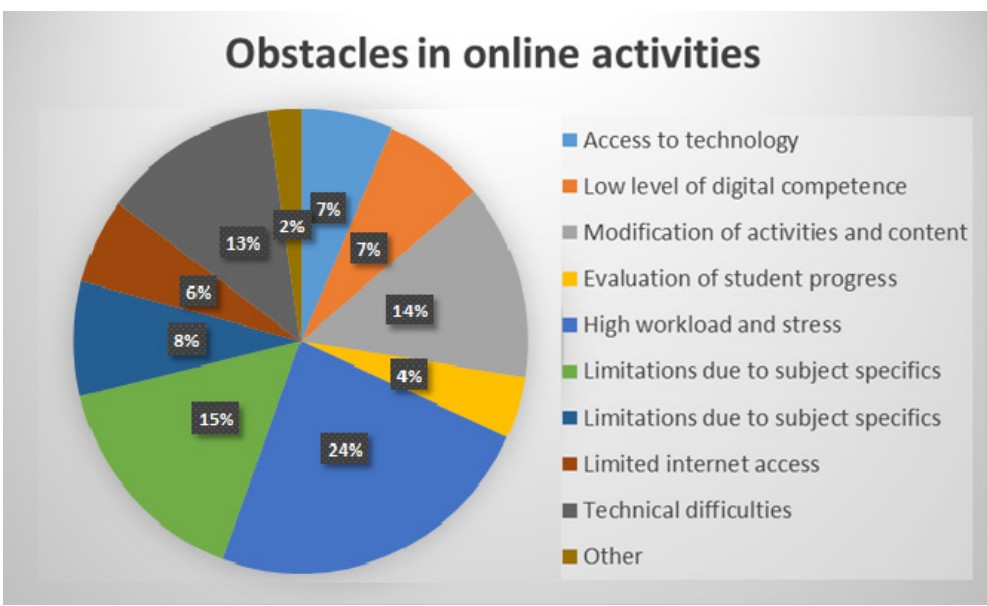

**Figure 12.** Obstacles in online activities.

Regarding the preparation of future engineers, most respondents concluded that they should focus on both technical knowledge and competences (Figure 13).

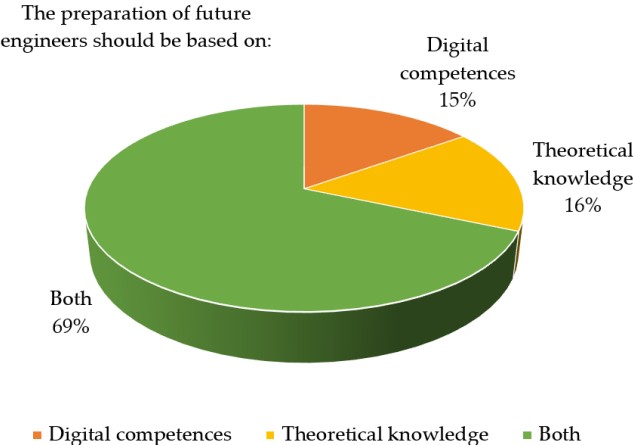

**Figure 13.** Main elements on which the preparation of future engineers should focus on.

## 4. Results

As a result of applying this questionnaire, some positive and negative effects of online education on professional training can be emphasised, as they are presented in Table 1:

**Table 1.** The main positive effects and negative effects on professional training.

| Positive Effects | Negative Effects |
|---|---|
| possibility of participation in activities for students who work; | lack of laboratory and practical internships, leading to insufficient understanding of the information, lack of skills in identifying, measuring, and testing equipment and difficulty in processing the results. In many cases, the laboratory activities focused on the theoretical aspects due to the impossibility of carrying out measurements or physical experiments; |
| gaining of time by eliminating dead time caused by travelling to teaching activities; | more time allocated for professional training, determined by more information content, many topics, and the need of a longer time for individual study; |
| development of digital skills and additional computer training and use of dedicated software determined by the need to use computers, applications, and online platforms; | lack of socialization with colleagues and teachers, which limits or hinders sharing of ideas, argumentation, justification, teamwork, etc.; |
| increasing students' responsibility by the requirement to complete tasks within a set deadline; | disruptions (distracting domestic stimuli) leading to poor concentration, increased stress, and lack of motivation; |
| independent learning, stimulating interest for access to more information, determined by the need for individual training, without the support and constant help of a teacher; | reduced impact of the message delivered online compared to face-to-face, sometimes due to insufficient training of some teachers for online teaching; |
| the possibility to review the taught material, as it was posted on the online learning platform, and the possibility to review the material, at any time, helped the student to understand the taught material correctly and quickly; | problems connecting and accessing online platforms (missing or malfunctioning/broken computer, inability to install specialised software, lack of internet access. |
| spending more time with family for students studying in a different location than their home. | |

Students' suggestions for improving the online teaching process were considered:

- More examples discussed by teachers, explanations should be punctual and clear, and students' level of preparation should be considered;
- Synthesize the taught material and reduce the volume and the number of homework assignments;
- Better training of the teachers for online activities to maintain students' attention by using different teaching platforms, creating more attractive courses;
- Improved interaction and communication between teachers and students;
- Video recording the experiments carried out in the labs, with step-by-step explanations of the realized actions;
- Design of special software, capable of providing remote access to laboratory stands for experiments;
- Shorten the duration of the lessons or combine them with more engaging activities that break the monotony and that are more attractive for students;
- Easy access to dedicated software, through the purchase of student licenses;
- Financial support for students and teachers in the purchase of devices necessary for online activities;
- Recommend more bibliographical resources for in-depth and effective documentation;
- Performing face-to-face laboratory activities to acquire practical skills needed at jobs, where physical interaction with the facility is required;
- Motivate students that are enthusiastically involved in the activities;
- clear presentation of the requirements, conditions, and criteria for student evaluation;
- All teachers from the same specialization to use the same online teaching platform.

## 5. Discussion

A study on 547 students and 213 teachers at a university in the Kingdom of Saudi Arabia highlighted the necessity of e-learning acceptance by all partakers involved in the educational process, as well as *the training of teachers for this teaching system correlated with the students, staff, and infrastructure* [6]. Another study conducted in UK, involving 555 students from The Open University, has indicated that 36% of respondents reported *negative impact on distance learning activities* (learning, evaluation, and social activities), and only 15% reported a positive impact [7]. In research performed in Afghanistan, regarding students from higher education, including 592 respondents who participated online in six semi-structured interviews, *it has been identified that the trainees did not benefit from a consistent teaching–learning experience* [8]. The most difficult technical issues concerned *internet connection* and the *access to expensive platforms*. In addition to those mentioned above, similar studies have also reported the negative impact determined by the *internet services quality* or availability, the impossibility of providing mobile devices for all children (especially in the case of families with several children), and the lack of concentration caused by the sharing of a single learning space by several siblings [9]. It has been noticed that, in developed countries, 95% of scholars have an available computer to use for schoolwork, while, in the underdeveloped countries, this percent is much lower. Another research study in China, which focused on higher education teachers' experiences during the COVID-19 pandemic, concluded that *online learning has limitations related to a lack of social interaction and case study*, but brings short- and long-term benefits, such as less anxiety and depression, which might arise due to social distancing and inactivity, as well as changing the traditional teaching pattern by integrating information and communication technology (ICT) into the teaching act, which should be a student-centred one.

The impact of the shift from face-to-face to online learning among students has also been examined in research that evaluated student motivation and learning performance [10]. In this analysis, 282 students participated and reported differences between the period before and during the COVID-19 restrictions. The research found that students *lost motivation and learning performance during the restrictions*. Students experienced an increase in stress, anxiety, fatigue, struggled to maintain concentration during online activities,

and had to sustain additional costs caused by using internet services and purchasing devices/software/platforms required for learning [11–14]. In another analysis performed on students from six Geography universities from three different countries, a lack of *practical experience, lecture interactivity, and adequate workspaces was identified* [15]. Furthermore, in the case of medical students, it was found that *practical skills are better formed and developed in clinics and laboratories, and, consequently, many students do not feel sufficiently prepared to sustain their graduation exams after the e-learning sessions* [16]. Although the study focused on another field, namely medicine, and the research was performed only on students of one university, it can be considered appropriate to be included in the current research, since the results highlight that online education is suitable for certain fields of activity. Thus, according to the above-mentioned research, only 24.5% of medical students, from the examined institution, were satisfied with online learning, and 50% of them reported *difficulties in staying motivated*, considering that online activities in medical higher education must be improved on three levels: *quality of applied pedagogy*, quality of course content, and preparation for the course.

The effects of the restriction in this period are related to a lack of communication and socialization, the long time spent in front of the computer, the need for supervising children during learning, the absence of discipline during classes, reduced innovation skills, and increased bills caused by additional use of internet services. To mitigate this impact, it is *necessary to provide students with specialised support*, so that the gap in online learning compared to face-to-face activities is eliminated [17]. Moreover, educators have identified *difficulties in implementing the content of the curriculum* due to limitations caused by using adapted learning strategies, the long time durations spent in front of the computer for lesson preparation, homework checking, and activities, and dead time due to waiting or poor internet connectivity. Teachers' approaches varied during the online activities. Many teachers preferred to reduce the amount of course content and homework for students. However, the time spent by students for learning was quite reduced [18].

Changes in teaching strategies, assessment criteria, and meanings of evaluation practices were also identified among some teachers, surveyed online, in a study conducted in Brazilian universities [19]. A fundamental issue in online teaching is defined by the *necessity of digital resources such as platforms, libraries, and adequate technical equipment and materials for guidance and study*. Equally important is also the psycho-emotional state and training of teachers involved in online teaching, since they are the planners of the learning process, and their training restructures both theoretical and practical learning [20–22]. In order to find solutions for adapting to online teaching, some researchers analysed the responses given by 487 students in Thailand, and, based on the developed models and modelling techniques, they discovered new learning strategies. Thus, after analysing students' opinions and perceptions on the whole system that contributes to learning, solutions for educational management adapted to the students' needs were proposed, which both educational institutions and policymakers should consider [23]. However, the transition from the traditional education system to the online systen was positively felt among students, since they could benefit from continuing education without having to travel to other locations and having no additional costs.

In a study on 103 young students aged between 20 and 23, which assessed several criteria for both distance and formal English language learning (academic performance, concentration and memory skills, progress in oral and written assignments, ability to absorb information, physical and mental health), some advantages were identified, such as: *increased free time*, the ability to take more breaks, the provision of a *more comfortable learning environment*, the lack of travel to the learning institution, and the absence of the severe control of the teacher [24]. The evaluation of students' increased academic performance through self-assessment can also be based on subjective evaluation, or through a less strict control of knowledge, which indicates that special attention should be paid to self-assessment. Better academic performance in online versus face-to-face learning was also identified in a study conducted in the United Arab Emirates [25]. The analysis found

that students who performed well in face-to-face learning had similar results in online learning, while students who performed poorly in face-to-face learning manifested 11% worse results in online learning. Switching courses to an online system was regarded as a satisfying experience, especially by students that work and intend to continue their studies simultaneously. Another similar study on 300 students concluded that the use of an e-learning platform is an effective method for developing technical and digital skills [26]. It can be stated that e-learning is considered a satisfactory tool in acquiring knowledge; however, it is *not sufficiently effective for acquiring clinical and technical skills*. Thus, after overcoming some of the technical challenges, e-learning can create new opportunities for the use of blended learning approaches.

Providing additional learning support can reduce the learning gap between students with different social backgrounds and with different development stages and where pre-existing social inequalities are identified [27]. Hence, through adequate support provided by the government and with an efficient learning system, 96% of Indonesian students felt comfortable and satisfied with the online learning experience [28]. At the same time, the most important aspects of staff training can be identified to develop a strategy to harmonize education and diversify the curriculum, including social interaction [15]. The employment of different learning methods can lead to an increase in the quality of the educational act and provide a friendly and beneficial climate in this process. Consequently, M-learning was considered in a study involving 116 students as an effective solution for online learning. The *factors that have been identified as being supportive for the education process were the use of interactive video materials, the ability to access information and online materials, appropriate assessment tools, mobile phone-compatible e-learning platforms, the ability to use different communication platforms (email, WhatsApp), and the ability to plan and organize time* [29,30].

In the field of engineering, it has been found that e-learning tools supported by full *access to technology* are able to enhance theoretical learning and the competences of the students. These aspects, correlated with the qualities of the teachers (effective communication skills, diversified teaching methods, competent use of technique and technology, flexibility, approachable, and supportive attitude towards teaching) can play a positive role in preparing students in the online environment [31]. The limitations of this research are determined by the fact that only one instrument was used, namely the questionnaire; therefore, future research should be considered, assessing the impact of the pandemic on ethnic education by using qualitative approaches such as interviews with, and focus groups of, engineering students. Moreover, in order to assess the impact of the pandemic in the long term, this questionnaire should be administered to the same subjects in the current academic year or in one year's time to identify any changes in students' perceptions.

Comparing the findings obtained and presented in the mentioned research with the results of the present research, a very high similarity was found. Thus, it was found that, during the period of online activities, students felt a lack of socialization, that they lost their motivation and concentration much more easily, and became anxious. Other problems identified in all the research reviewed concerned a lack of access or difficult access to technology, technical problems of connection, and the need for adaptation and preparation of both the teachers and students for online learning activities. Concerning the applied activities (laboratory, practical) in all the research analysed, students expressed dissatisfaction with the online activities, which are not useful in all cases for the formation of skills and competences.

Regarding the positive aspects identified, it was found that, in all the cases analysed, students felt that they acquired more digital skills, benefited from a comfortable learning environment, and gained more time by eliminating travel and thus reducing expenses.

Compared to the studies analysed in this study, the participation of students in all activities was also analysed, and 72% of respondents participated. Moreover, from the analysis, it was identified that students prefer face-to-face activities; however, at the same time, they consider it useful to keep some activities online.

Regarding the assessment mode in the studio, it was identified that students prefer to be checked as they go along, especially through grid tests.

As internet services in Romania are not so expensive in this study, this impediment was not identified. Another barrier identified only in this study is the large amount of time allocated by students to participate in activities and to prepare assignments.

The findings from the study largely confirm the conclusions identified in other similar work, but also provide an opportunity to identify solutions specific to the country in which the research is conducted, given that each nation may or may not benefit from certain facilities.

## 6. Conclusions

The conclusion drawn from this study is that e-learning could be a suitable method for teaching and learning in technical faculties, though as a complementary activity to face-to-face teaching. Some students prefer face-to-face activities, considering the theoretical knowledge and competences more relevant than digital skills. As far as we are concerned, we believe that the various digital skills acquired during online teaching activities will be useful to students in their professional work, being in line with the trend of increasing technology and adaptation to the careers of the future. However, a number of drawbacks have been identified in online activities, and many of these have been overcome, given the pass rate of the exams during the pandemic period.

Online teaching is and remains a replacement for direct communication, a second-choice substitute, with the advantage of using educational platforms that link people interested in a particular subject, even if they are in different places of a country or even of the world, but with multiple disadvantages linked to the quality of the teaching process. It is likely that, in the future, the solution will be to hybridize higher education, and, in Romania, there are also legislative initiatives in this regard; however, this should be only done for courses whose content can be transmitted through this communication channel. Therefore, in a possible hybrid education system, students' suggestions should be considered based on the shortcomings they faced during the pandemic period, as well as the harmful effects in the long term. Education, and the ways in which it is delivered, is responsible for the development of the individual and whole society.

Future studies could also look at a comparative analysis of the skills of students from technical faculties who took courses before and after the pandemic, and possibly employer satisfaction with graduates of technical studies who took only face-to-face courses and those who participate in a hybrid education.

**Author Contributions:** Conceptualization, E.S. and N.-M.F.; methodology, E.S. and G.M.; validation, E.S. and N.-M.F.; formal analysis, E.S. and N.-M.F.; investigation, E.S., N.-M.F., R.M. and G.M.; resources, E.S., N.-M.F., R.M. and G.M.; data curation, E.S.; writing—original draft preparation, E.S. and G.M.; writing—review and editing, N.-M.F. and G.M.; visualization, N.-M.F. and R.M.; supervision, E.S.; project administration, E.S.; funding acquisition, E.S., N.-M.F. and R.M. All authors have read and agreed to the published version of the manuscript.

**Funding:** The publication of this article was supported by the 2021 Development Fund of the UBB. And the APC was partially funded by the Westphalian University of Applied Sciences Gelsenkirchen Bocholt Recklinghausen through the Open Access Publications Fund.

**Institutional Review Board Statement:** Not applicable.

**Informed Consent Statement:** Not applicable.

**Data Availability Statement:** The data presented in this study are available on request from the first author.

**Conflicts of Interest:** The authors declare no conflict of interest. The funders had no role in the design of the study; in the collection, analyses, or interpretation of data; in the writing of the manuscript, or in the decision to publish the results.

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
