# Peer review of "Impact of COVID-19 Pandemic on the Educational-Instructional Process of the Students from Technical Faculties"

_sustainability, doi:10.3390/su14148586_

Round 1

Reviewer 1 Report

Comments and Suggestions for Authors

In the present work, Elisabeta Spunei et al. try to explain the impact of COVID-19 pandemic on the educational-instructional process of the students from technical faculties. There are some questions that should be considered.

1. ‘Introduction’ is too long, it should be revised thoroughly. Hypothesis and objectives should be included in ‘Introduction’. There are many long sentences that are difficult to understand.

2. ‘Materials and Methods’ should be revised. ‘The research objectives’ should be included in ‘Introduction’.

3. ‘3. Descriptive analysis of questionnaire results’ is not suitable for a subtitle. A subsection of Statistical analyses and Result section are needed.

4. The main positive effects and negative effects in ‘Discussion’ should be included in Result section, which may be present as two Tables.

5. There is no discussion in Discussion section. There are too contents in Introduction and Conclusions sections, some contents may be moved to Discussion section.

6. This study is involved in human, and Ethics approval should be included.

7. Format of references and authors address is not suitable for this Journal.

8. Editing of English language and style is needed.

For example, long sentence “The general conclusion drawn from the analysis was that, although the pandemic has had adverse consequences on health and life quality of many people, the challenges that humankind has been subjected to, have led to personal and professional development of many people, and opened up new perspectives for carrying out the everyday activities.”

Lines 27-27, “in the way people live”.

Line 167, “between March 2021 and July 2021 The study”

Author Response

Dear Reviewer,

Thank you for your careful analysis of our work. We have taken into account your suggestions and recommendations.

Comments and Suggestions for Authors

In the present work, Elisabeta Spunei et al. try to explain the impact of COVID-19 pandemic on the educational-instructional process of the students from technical faculties. There are some questions that should be considered.

  1. ‘Introduction’ is too long, it should be revised thoroughly. Hypothesis and objectives should be included in ‘Introduction’. There are many long sentences that are difficult to understand.

Given your recommendation we have made a revision of the introduction to reduce it.

  1. ‘Materials and Methods’ should be revised. ‘The research objectives’ should be included in ‘Introduction’.

The research objectives and hypotheses were included in the introduction.

  1. ‘3. Descriptive analysis of questionnaire results’ is not suitable for a subtitle. A subsection of Statistical analyses and Result section are needed.

For a better understanding of the results obtained from the research, we considered it appropriate to keep this section in the form of Statistical Analysis.

  1. The main positive effects and negative effects in ‘Discussion’ should be included in Result section, which may be present as two Tables.

The positive and negative aspects initially presented in the Discussion section have been formulated in the Results section.

  1. There is no discussion in Discussion section. There are too contents in Introduction and Conclusions sections, some contents may be moved to Discussion section.

The Discussion section has been reworded in its entirety.

  1. This study is involved in human, and Ethics approval should be included.

The questionnaire applied was anonymous, the respondent is not identifiable, thus the information collected is not considered personal data. Anonymization is irreversible.

  1. Format of references and authors address is not suitable for this Journal.

We have revised the format of the references.

  1. Editing of English language and style is needed.

For example, long sentence “The general conclusion drawn from the analysis was that, although the pandemic has had adverse consequences on health and life quality of many people, the challenges that humankind has been subjected to, have led to personal and professional development of many people, and opened up new perspectives for carrying out the everyday activities.”

Lines 27-27, “in the way people live”.

Line 167, “between March 2021 and July 2021 The study”

We revised the English translation and reworded the identified phrases.

Reviewer 2 Report

This is an interesting paper about student’s experiences of instruction during the period of the Covid-19 pandemic and the pivot to remote teaching. The focus is specifically on students from technology faculties across 9 universities, with responses from 249 students. I think the findings are interesting and useful, and it seems possible that a revised version of this paper might be publishable in the journal.

However, the present draft presents a number of important weaknesses that, in my view, should be address by the authors in a revised version before publication is considered:

  • The Introduction needs to give more of an overview of the paper and the present work, rather than leaving this information to be discovered by the readers in later sections.

  • The themes drawn out from prior literature on pp. 1-4 need to be made more explicit, perhaps by the use of italics. The authors correctly highlight a focus on issues such as e-learning acceptance, student motivation teaching strategy development, etc., but these could be signposted more clearly. Once this is done, the key point will be to emphasise what is missing in the present literature, such that the present work might seek to make a contribution.

  • I think the Introduction should avoid referring to papers of dubious relevance, for example the work on schools towards the bottom of p. 2. The review could be slightly shorter and more to the point if this is done.

  • The final paragraph of the Introduction should be strengthened by signposting the section structure of the rest of the paper.

  • Section 2 lines 165-179 could be strengthened by arguing how this work adds something new to the literature.

  • Where explaining the research objectives, I think we need more of a sense of WHY these objectives were chosen, rather than just stating what the objectives were.

  • A similar justification should be provided when establishing the detailed lists of “causes” and other items included in the questionnaire: why were these chosen? Were they derived from prior literature, the authors’ experiences, or somewhere else?

  • Section 3 could be strengthened by explaining the structure in advance, and by stating explicitly how this structure is related to the research objectives stated earlier. I think that the introduction of sub-headings would also be useful.

  • I think the Discussion section is the weakest section of the paper, which would benefit from significant re-writing. The key point is to identify how the findings contribute something new to the literature (the same literature you reviewed earlier) in a way that other scholars would find interesting. At present that argument is not attempted, and in my view the section should be re-written in order to do so.

  • The Conclusion might better state the core findings for other researchers, and establish what future work is suggested by the present findings.

Author Response

Dear Reviewer,

Thank you for your careful analysis of our work. We have taken into account your suggestions and recommendations.

This is an interesting paper about student’s experiences of instruction during the period of the Covid-19 pandemic and the pivot to remote teaching. The focus is specifically on students from technology faculties across 9 universities, with responses from 249 students. I think the findings are interesting and useful, and it seems possible that a revised version of this paper might be publishable in the journal.

However, the present draft presents a number of important weaknesses that, in my view, should be address by the authors in a revised version before publication is considered:

The Introduction needs to give more of an overview of the paper and the present work, rather than leaving this information to be discovered by the readers in later sections.

Given your recommendation we have made a revision of the introduction to reduce it.

The themes drawn out from prior literature on pp. 1-4 need to be made more explicit, perhaps by the use of italics. The authors correctly highlight a focus on issues such as e-learning acceptance, student motivation teaching strategy development, etc., but these could be signposted more clearly. Once this is done, the key point will be to emphasise what is missing in the present literature, such that the present work might seek to make a contribution.

References to existing results in the literature have been introduced in the discussion section, highlighting the significatory aspects, an aspect also found in our research.

I think the Introduction should avoid referring to papers of dubious relevance, for example the work on schools towards the bottom of p. 2. The review could be slightly shorter and more to the point if this is done.

I have removed references to the results of research that focused on another age group, including the bibliographic reference you refer to.

The final paragraph of the Introduction should be strengthened by signposting the section structure of the rest of the paper.

We have introduced the following clarifications:

The paper is structured in four sections in which the methods used in the research, the statistical analysis of the primed responses and the results obtained sited in the form of positive and negative effects on the teaching process, and the discussion section are presented.

Section 2 lines 165-179 could be strengthened by arguing how this work adds something new to the literature.

The research carried out so far provides general information, without specifically analysing the impact of e-learning on the teaching process in the technical field, where application activities in specialized laboratories must be predominant.

Where explaining the research objectives, I think we need more of a sense of WHY these objectives were chosen, rather than just stating what the objectives were.

The objectives of the research were set based on the research carried out up to the date of the questionnaire but especially on the specifics of the teaching process in the technical field.

A similar justification should be provided when establishing the detailed lists of “causes” and other items included in the questionnaire: why were these chosen? Were they derived from prior literature, the authors’ experiences, or somewhere else?

The covered items and the answer variants included in the questionnaire were formulated on the basis of consultation with teachers in the field who have faced different problematic situations with the transition to the online system.

Section 3 could be strengthened by explaining the structure in advance, and by stating explicitly how this structure is related to the research objectives stated earlier. I think that the introduction of sub-headings would also be useful.

I think the Discussion section is the weakest section of the paper, which would benefit from significant re-writing. The key point is to identify how the findings contribute something new to the literature (the same literature you reviewed earlier) in a way that other scholars would find interesting. At present that argument is not attempted, and in my view the section should be re-written in order to do so.

We have restructured the paper, in the discussion section references have been made to existing findings in the literature on learning issues during a pandemic.

The Conclusion might better state the core findings for other researchers, and establish what future work is suggested by the present findings.

Throughout the discussion section we have highlighted in italics aspects contained in the literature and confirmed by this research.

Reviewer 3 Report

The aim of this study was to identify the impact of students’ educational experiences, when switching from face-to-face to online education, during public health emergency or Covid 19 - related state of alert. This article is timely and generally well written. A number of issues arise from reading it and are reported below.

1/ In the method section, can you justify the university chosen to conduct the study?

2/ How did you mobilise the list of students for your questionnaire?

3/ Was a CNIL declaration (data protection) made or ethics committee? 

4/ Could you please specify the software used to distribute the questionnaire.

5/ Justify sample size.

6/ In the result section, can you specify the number of reminders and the response rate to this questionnaire.

7/ In the discussion section, you should compare your results to the literature and not repeat your results in the discussion section. This section needs to be improved.

8/ The authors introduced a “limitation” in the conclusion. I suggested to introduced this section in the discussion chapter.

Author Response

Dear Reviewer,

Thank you for your careful analysis of our work. We have taken into account your suggestions and recommendations.

The aim of this study was to identify the impact of students’ educational experiences, when switching from face-to-face to online education, during public health emergency or Covid 19 - related state of alert. This article is timely and generally well written. A number of issues arise from reading it and are reported below.

  1. In the method section, can you justify the university chosen to conduct the study?

In carrying out this research, not a specific university was selected, but a fundamental field of engineering, given that the curricula include application activities that require the use of specific equipment, to which students could not have access during online activities.

  1. How did you mobilise the list of students for your questionnaire?

The questionnaire was distributed through university staff and collaborators from other university centres. Students were given the option to complete the questionnaire, there was no obligation to complete it, they were interested in completing the questionnaire so that they could observe the problems they faced during the pandemic.

  1. Was a CNIL declaration (data protection) made or ethics committee?

The questionnaire applied was anonymous, the respondent is not identifiable, thus the information collected is not considered personal data. Anonymization is irreversible.

  1. Could you please specify the software used to distribute the questionnaire.

The questionnaire was developed using a Google tool (Google Forms), which allows you to conduct surveys and centralize the results in an Excel spreadsheet. The authors have access to these tools under institutional licenses.

  1. Justify sample size.

In the sample we started from an average of existing students in the 9 faculties, i.e. a population of 150,000 students. Taking into account a confidence coefficient of 95% and a margin of error of 7%, we have determined a representative sample of 197 respondents. Following the free distribution of the questionnaire we collected 224 responses, which leads us to state that the results are representative for the selected population.

  1. In the result section, can you specify the number of reminders and the response rate to this questionnaire.

The questionnaire was distributed through teachers' communication channels and was open for completion throughout the period. No reminders were made during the data collection period.

  1. In the discussion section, you should compare your results to the literature and not repeat your results in the discussion section. This section needs to be improved.

We have restructured the paper, in the discussion section references have been made to existing findings in the literature on learning issues during a pandemic.

  1. The authors introduced a “limitation” in the conclusion. I suggested to introduced this section in the discussion chapter.

The limitation originally presented in the conclusions has been moved to the discussion section, as per your suggestion.

Round 2

Reviewer 1 Report

In the revised manuscript, authors try to revise this manuscript thoroughly. However, there are some questions that should be considered.

1. Format of authors address and references is not suitable for this Journal in revision.

2. In ‘Introduction’, there is no reference.

3. Format of writing is wrong in ‘Introduction’. There are ‘1. Introduction’, ‘1. to assess the impact of the measures….’, ‘2. the students’ adaptability….’ and ‘3. the evaluation of the…’.

4. In ‘Statistical analysis’ section, authors should write the methods of statistical analysis used in this study, not some results.

5. In ‘Discussion’ section, authors should discuss and compare with your findings.

Author Response

Dear Reviewer,

Thank you for your careful analysis of our work. We have taken into account your suggestions and recommendations.

Comments and Suggestions for Authors

In the revised manuscript, authors try to revise this manuscript thoroughly. However, there are some questions that should be considered.

Format of authors address and references is not suitable for this Journal in revision.

We have revised the authors' address and bibliographic references

 In ‘Introduction’, there is no reference.

 We have inserted 5 bibliographical references in the introduction

 Format of writing is wrong in ‘Introduction’. There are ‘1. Introduction’, ‘1. to assess the impact of the measures….’, ‘2. the students’ adaptability….’ and ‘3. the evaluation of the…’.

 We have revised the form of the wording in the introduction, replacing 1..., 2..., 3 with A ......., B ........., C.......

 In ‘Statistical analysis’ section, authors should write the methods of statistical analysis used in this study, not some results.

We described how We proceeded to process the students' responses to the questions asked:

After closing the questionnaire developed in Google Forms we used the Spreadsheet tool, which allowed us to centralize all the answers given by the students to the questions asked. Based on the answers obtained, we drew up graphs, which were intended to make it easy to highlight the results obtained. For the open questions, we grouped the answers roughly similar in terms of theme, resulting in the main positive and negative aspects identified in the professional training during the online activities.

 In ‘Discussion’ section, authors should discuss and compare with your findings.

We discussed the results obtained compared to those mentioned in other scientific papers:

Comparing the findings obtained and presented in the mentioned researches with the results of the present research, a very high similarity was found. Thus, it was found that during the period of online activities students felt the lack of socialization, lost their motivation, concentration much more easily and became anxious. Other problems identified in all the research reviewed concerned lack of access or difficult access to technology, technical problems of connection and the need for adaptation and preparation of both teachers and students for online learning activities. Concerning the applied activities (laboratory, practical) in all the research analysed, students expressed dissatisfaction with the online activities, which are not useful in all cases for the formation of skills and competences.

Regarding the positive aspects identified, it was found that in all the cases analysed students felt that they acquired more digital skills, benefited from a comfortable learning environment, gained more time by eliminating travel and thus reducing expenses.

Compared to the studies analysed in this study, the participation of students in all activities was also analysed and 72% of respondents participated. Also from the analysis it was identified that students prefer face to face activities, but at the same time consider it useful to keep some activities online.

Regarding the assessment mode in the studio, it was identified that students prefer to be checked as they go along, especially through grid tests.

As internet services in Romania are not so expensive in this study this impediment was not identified. Another barrier identified only in this study is the large amount of time allocated by students to participate in activities and to prepare assignments.

The findings from the study largely confirm the conclusions identified in other similar work, but also provide an opportunity to identify solutions specific to the country in which the research is conducted, given that each nation may or may not benefit from certain facilities.

Reviewer 3 Report

The authors have taken my remarks into account.

Author Response

Dear Reviewer,

Thank you for your appreciation and for taking the time to review our work. All the best,

Round 3

Reviewer 1 Report

Comments and Suggestions for Authors

Thanks for addressing some review comments. However, it is not suitable for publication.

1. In ‘Statistical analysis’ section, authors should write the methods of statistical analysis used in this study, not some results. ‘Statistical analysis’ is only a subsection of Materials and Methods section.

2. There are so many paragraphs that only include several sentences.

3. The main positive effects and negative effects may be present in the formation of tables.

4. The writing ability of authors is poor. The manuscript should be revised throughout by a scientific expert.

Author Response

Impact of COVID-19 pandemic on the educational-instructional process of the students from technical faculties

Dear Reviewer,

Thank you for your careful analysis of our work. We have taken into account your suggestions and recommendations.

  1. In ‘Statistical analysis’ section, authors should write the methods of statistical analysis used in this study, not some results. ‘Statistical analysis’ is only a subsection of Materials and Methods section.

We have completed the section with details of the statistical analysis methods used.

  1. There are so many paragraphs that only include several sentences.

We filled in the paragraphs we identified in this way, adding further details or restructuring the information. Initially I inserted a paragraph after each figure, considering that this made the work easier to understand.

  1. The main positive effects and negative effects may be present in the formation of tables.

In accordance with your recommendation, we have presented, the positives and negatives effects on professional training, in the formation of tables.

  1. The writing ability of authors is poor. The manuscript should be revised throughout by a scientific expert.

In accordance with your recommendation, the entire work has been revised by a certified English translator and changes have been made where necessary.

We thank you and kindly ask you to review our paper, as we are very eager to publish the results of our research.

Round 4

Reviewer 1 Report

Comments and Suggestions for Authors

Thanks for addressing some review comments. However, it is not suitable for publication.

1. In ‘Statistical analysis’ section, authors should write the methods of statistical analysis used in this study, not some results. ‘Statistical analysis’ is only a subsection of Materials and Methods section. This has been suggested at second and third revision required. However, it has not revised.

2. In the ‘Introduction’ section, there are different serial numbers (A., (a) and b)).

3. The writing ability of authors is poor. The manuscript should be revised throughout by a SCIENTIFIC expert, not an editor of English language.

Author Response

Hello dear reviewer,
In the full version we have tried to take into account your comments and those of the other reviewers who have already agreed to publication.
In the expanded version we have also taken into account the comments of the editor.
Thank you,
